# Importance of Synthesizing High-quality Data for Text-to-SQL Parsing

**Yiyun Zhao**[*]
University of Arizona

**Jiarong Jiang, Yiqun Hu, Wuwei Lan, Henry Zhu, Anuj Chauhan, Alexander Li,
Lin Pan, Jun Wang, Chung-Wei Hang, Sheng Zhang, Marvin Dong, Joe Lilien,
Patrick Ng, Zhiguo Wang, Vittorio Castelli, Bing Xiang**
AWS AI Labs
{jiarongj, yiqunhu, lanwuwei, patricng}@amazon.com

## Abstract

There has been increasing interest in synthesizing data to improve downstream text-to-SQL tasks. In this paper, we examined the existing synthesized datasets and discovered that state-of-the-art text-to-SQL algorithms did not further improve on popular benchmarks when trained with augmented synthetic data. We observed two shortcomings: illogical synthetic SQL queries from independent column sampling and arbitrary table joins. To address these issues, we propose a novel synthesis framework that incorporates key relationships from schema, imposes strong typing, and conducts schema-distance-weighted column sampling. We also adopt an intermediate representation (IR) for the SQL-to-text task to further improve the quality of the generated natural language questions. When existing powerful semantic parsers are pre-finetuned on our high-quality synthesized data, these models have significant accuracy boosts and achieve new state-of-the-art performance on Spider.

## 1 Introduction

Text-to-SQL semantic parsing translates a natural language question (NLQ) to a corresponding SQL query. In recent decades, many industries have adopted high-level digitalization in their workflow and possessed large-scale datasets—many of which are stored as relational databases. Extracting insights from these relation databases to further drive business decisions is an important task. But due to the complexity of these relational databases, query language experts are often needed to extract valuable insights. Thus a high-performing text-to-SQL system with a natural language interface would greatly lower the barrier for users to query their databases.

In order to obtain high-quality training data for the text-to-SQL parser, human annotators with SQL expertise are needed to construct NLQ-SQL parallel data, which are difficult and expensive to scale. Thus data scarcity is a well-known bottleneck in the text-to-SQL task [Yu et al., 2018b]. To address the data scarcity issue, there is an increasing interest in leveraging synthetic data to improve downstream performance. Yu et al. [2021] handcrafted high-quality rules to synthesize SQLs and NLQs simultaneously, but these grammar rules need to be carefully designed through expensive manual work. To automate the synthesis procedure, recent studies [Wang et al., 2021, Wu et al., 2021, Shi et al., 2021, Zhong et al., 2020] utilize a two-stage approach that synthesizes SQLs first and then composes NLQs with a SQL-to-text generator. Alternatively, Yang et al. [2021] proposed a

---

[*]Work done during an internship at AWS AI Labs.

NeurIPS 2022 Workshop on Synthetic Data for Empowering ML Research.

reversed pipeline that uses an entity-to-question model to generate natural language queries and then a text-to-SQL parser to generate SQL queries.

In this paper, we delve into the two-stage synthesizing method that first synthesizes SQL queries and then generates NLQs. We propose a novel framework[2] that has several strategies to reduce synthesis errors present in existing methods. During the stage of SQL synthesis, we employ template synthesis with strong typing, template key relationship preservation, and schema-distance-weighted column sampling. During the stage of text generation, we propose an intermediate representation to bridge the gap between SQL queries and natural language questions. We show that models trained with our synthetic datasets outperform the models trained with previous synthetic datasets. Our model achieves new state-of-the-art accuracy on the Spider benchmark.

In summary, our main contributions are:

- We systematically compare the existing text-to-SQL synthesis methods and identify three root causes of low quality;

- We propose several novel strategies to data synthesis and demonstrate augmentation benefits when using the state-of-the-art PICARD parser, underscoring the importance of the synthesis quality;

- We adopt an intermediate representation (IR) for the SQL-to-text task, which can further improve the quality of the generated natural language questions.

## 2 Existing Synthesis Methods and Limitations

We conduct a detailed investigation towards the existing text-to-SQL synthesis methods to understand each of their advantages and shortcomings, the details of which can be found in Appendix A. In particular, Figure 3 summarizes and compares the key characteristics from different dimensions.

In this section, we first experimented with two recent synthetic datasets [Wang et al., 2021] and [Wu et al., 2021] using the latest state-of-the-art text-to-SQL model PICARD [Scholak et al., 2021] to assess their effectiveness and found that these two recent synthetic datasets show only negligible impact on downstream accuracy when trained on the PICARD model in a data augmentation fashion. We then discuss the three main issues in these synthetic datasets based our manual inspection.

### 2.1 Synthetic Data Effectiveness Assessment

As a pilot study, we use T5-Large PICARD as the baseline parser to examine the synthetic data quality. As shown in Figure 1, the exact match (EM) accuracy on both synthetic datasets are less than 0.2 during Stage 1 (trained with synthetic data only), in contrast to 0.6 with Spider training data only. This gap indicates the limited transferability from existing synthetic data to real data. Further finetuning on Spider training data in Stage 2 does not improve the baseline model. However, our synthetic data (IR2NLQ and SQL2NLQ) show better performance on these two stages. In the next sections, we reveal the synthetic data problems and detail our proposed method.

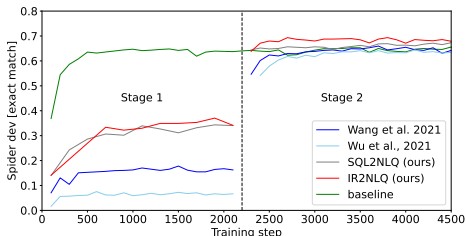

Figure 1: Training dynamics comparison of T5-Large with different synthetic data. The baseline model uses Spider real data only. IR2NLQ and SQL2NLQ are our synthetic data with and without IR during NLQ generation. We compare with previous synthetic datasets [Wu et al., 2021, Wang et al., 2021]. We use synthetic data for stage-1 training and real data for stage-2 training.

---

[2]https://github.com/awslabs/text2sql-ship

## 2.2 Synthetic Data Quality Analysis

We analyzed the previous synthesis methods to identify a few probable causes for obsolescence.

**Illogical SQL from invalid grammars.** The CFG designed by Wu et al. [2021] is constrained and they limited SQL generation to one table. While Wang et al. [2021] designed flexible grammars, they neglected the constraints between operators and column types. This neglect leads to mistakes such as `SUM(student.name)`, where an aggregation operator is applied to a text column.

Furthermore, PCFG generated SQL queries often failed to capture foreign-key and key relations between columns. This leads to invalid SQLs such as `SELECT name, age FROM student INTERSECT SELECT address FROM teacher`, where it intersects two sub-queries with different number of columns. In fact, designing a grammar to produce high coverage and logical SQLs is a difficult task due to the implicit dependencies of SQL elements.

**Over-Complex SQL from arbitrary joins.** When SQLs are materialized, the column/table selection from existing work is independent and result in SQL queries with unnecessary complexity. Those queries often have unclear intent and thus are difficult to be correctly translated to natural language questions. For instance, a simple template in Table 2 that requires only two columns can be turned into a complicated and nonsensical SQL query with three table joins.

**Language gap between SQL and NLQ.** Recent work typically trains a sequence-to-sequence model to obtain corresponding natural language queries (NLQ) from synthetic SQLs [Wang et al., 2021, Shi et al., 2021]. The gap between SQL-NLQ pairs are well recognized in text-to-SQL task, and intermediate representation (IR) is commonly used to reduce such mismatch [Gan et al., 2021b, Guo et al., 2019a, Yu et al., 2018a, Shi et al., 2021]. However, the reverse of the source and target in SQL-to-text brings in its own challenge, such as incorrect references for `SELECT *`, missing conditions within long and complex SQL queries, and misinterpretation of ORDER phrases.

## 3 Proposed Method

This section outlines our proposed synthesis pipeline (Figure 2). We follow the template based SQL synthesis approach similar to Zhong et al. [2020], Zhang et al. [2019] and generate corresponding NLQs with a sequence-to-sequence model. We address the generation problems reviewed in the previous sections by 1) introducing strong typing and encoding the key relation in templates for more logical SQLs; 2) proposing a schema distance weighted column sampling strategy to avoid over-complex joins; and 3) an improved IR to bridge the gap between SQL and NL questions specifically for SQL-to-text.

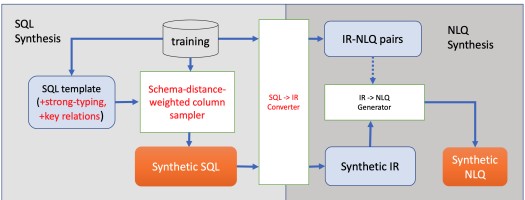

Figure 2: Our NLQ-SQL synthesis framework. Novel components include strong-typing, key relations, schema-distance-weighted column sampler, and SQL → IR converter.

### 3.1 SQL Synthesis

To create new SQLs on training data schemas, we utilize a template-based approach following Zhong et al. [2020]: First, a pool of SQL templates are created by normalizing the schema-related mentions (column and value) and removing JOIN phrases. During SQL generation, a template is sampled based on the training distribution and columns are sampled with constraints to fill in the normalized slots of the template. We highlight several improvement made to the existing approach.

**Strong Typing.** When normalizing columns, we enforce strong typing of a template by enriching and preserving the data type (e.g., text, number, date, etc) as well as key identity (key or not) for each column. For example, in Table 1, we use `textkey` instead of `key` to normalize `artist_name` because operators such as `MAX` can be applied to number key but usually not to other text key.

Table 1: Our modifications for template extraction: strong typing is highlighted in blue and key relation preservation is highlighted in pink.

| SQL | `SELECT artist_name FROM song INTERSECT SELECT artist_name FROM artist` |
|---|---|
| **Previous** | `SELECT col1_key INTERSECT col2_key` |
| **Ours** | `SELECT col1_textkey INTERSECT col2_textkey_fk1` |

**Template Key Relationship Preservation.** A foreign key is a column in a table referring to the primary key (unique identifier) of another table. In multiple table join scenarios, key and foreign key are the most common columns to be joined on. Restricting a column to be a foreign key to another key column is critical for a SQL to be valid especially in the following two cases: 1) queries including `INTERSECT`, `EXCEPT`, `UNION` and 2) queries that contains nested queries in `WHERE` conditions. For instance, the query in Table 1 implied the constraint that `song.artist_name` should be a subset of `artist.artist_name`. FK1 in the template captures the constraint of key relationship between the two `artist_name` columns, which prevents the template from generating nonsensical queries such as `SELECT gender FROM artist INTERSECT SELECT country FROM artist`.

**Schema-distance-weighted Column Sampling.** To mitigate the issue of arbitrary multi-table joins, we implement a weighted sampling function biased toward columns that are close, in terms of table distance, to the columns already selected in a SQL template.

For a given database $d$, we first establish an undirected graph for all the tables in $d$. Each table represents a node in the graph. The distance between any two tables, $e(\cdot, \cdot)$, is the least number of joins necessary to join the two tables (i.e. shortest path distance) under the restriction that table join can only take place with qualified primary key and foreign key pairs.

We design the schema-weighted column sampling algorithm (Algorithm 1 in Appendix B), which utilizes the table distances to control the column sampling weights. In particular, to sample a column $c$, we define the sampling weights for all other columns $w(\tilde{c}) = \begin{cases} 1, & \text{if } T_{\tilde{c}} = T_c \\ \frac{1}{\gamma^{e(c,\tilde{c})}}, & \text{o.w.} \end{cases}$, where $T_c$ denotes the table that column $c$ is in. The discussion on how to choose $\gamma$ can also be found in Appendix B. Such implementation is motivated from the observation that over-lengthy SQLs resulted from multiple tables joins are rare in real world scenarios under the only-join-on-primary-key-foreign-key assumption. Table 2 shows an example of how adopting the schema-weighted sampling can help reduce the unrealistic SQLs in the random case.

Table 2: Random sampling vs our schema-distance-weighted column sampling for a given template. The former produced a query with three joins while ours have both columns from the same table.

| **Template** | `SELECT col1_numberkey WHERE col2_name = VALUE` |
|---|---|
| **Random** | `SELECT T1.Club_ID FROM club AS T1 JOIN coach as T2 ON T1.Club_ID = T2.Club_ID JOIN player_coach AS T3 ON T2.Coach_ID = T3.Coach_ID JOIN player AS T4 on T3.Player_ID = T4.Player_ID where T4.Rank = "3rd"` |
| **Ours** | `SELECT Club_ID FROM club WHERE Club_Name="AIK"` |

## 3.2 NLQ Synthesis

Intermediate representation (IR) has been employed to simplify the SQL query with minimum information loss [Gan et al., 2021a, Guo et al., 2019b, Gan et al., 2021b, Guo et al., 2019a, Yu et al., 2018a, Shi et al., 2021]. Common operations include removing `FROM/JOIN` clauses and `GROUP BY` clauses, and merging `WHERE` clauses and `HAVING` clauses. Previous works find the use of IR often improves text-to-SQL performance.

In this section, we explore whether the SQL-to-text generation could also benefit from an IR. According to a prior research by Wu et al. [2021], altering the query's linearization order could already affect the synthetic text quality. The objective of an IR here is to convert SQL to a representation that more closely resembles the NLQ. This conversion involves both simplifications (such as removal of redundant information) and specification (such as introducing information using heuristics). In addition to the traditional designs,we introduced several additional rules to transform from SQLs to IRs, as listed below (examples in Table 3):

- Only drop tables in the `FROM/JOIN` phrase if they appear in other SQL elements (**EX2-EX4**). Removal of tables can simplify queries but tables in `JOIN` can also behave as filters and need to be preserved to avoid information loss (**EX1**).

Table 3: IR examples that illustrate the examples of removing tables, enriching * columns, specifying most/least intent, removing redundant `GROUP BY`. Unwanted intents are in grey, redundant intents are in green. Texts related to IR operations are highlighted with yellow.

| | | |
|---|---|---|
| **EX1** | SQL | `SELECT T1.name FROM student AS T1 JOIN has_pet AS T2 ON T1.student_id = T2.has_pet.student_id` |
| | IR | `SELECT name of student FROM has_pet` |
| | NLQ | Find the name of students who have pets. |
| **EX2** | SQL | `SELECT T2.name, count(*) FROM concert AS T1 JOIN stadium AS T2 ON T1.stadium_id = T2.stadium_id GROUP BY T1.stadium_id` |
| | IR | `SELECT name of stadium, Count ( record of concert ) GROUP BY ( stadium_id of concert )` |
| | NLQ | Show the stadium name and the number of concerts in each stadium. |
| **EX3** | SQL | `SELECT T1.neighbourhood_name neighbourhood AS T1 JOIN business AS T2 ON T1.business_id = T2.business_id WHERE T2.city = "Madison" GROUP BY T1.neighbourhood_name ORDER BY COUNT ( DISTINCE T2.name ) DESC LIMIT 1` |
| | IR | `SELECT neighbourhood_name of neighbourhood WITH most Count ( DISTINCT name of business ) WHERE city of business = "Madison"` |
| | NLQ | Which neighbourhood has the most number of businesses in Madison? |
| **EX4** | SQL | `SELECT T2.name FROM USER AS T2 JOIN review AS T1 ON T2.user_id = T1.user_id GROUP BY T2.name HAVING AVG (T1.rating) < 3` |
| | IR | `SELECT EACH ( name of user ) WITH Avg (rating of review ) < 3` |
| | NLQ | Find users whose average review rating is below 3. |

- Replace * in `count(*)` with the table whose columns in `JOIN` act as foreign key to provide explicit context for counting. This is because, in multi-table join queries, foreign key represents the **many** of the one-to-many relations and thus the rows from the table is more meaningful to be aggregated (see **EX2** replaces*with `concert` rather than `stadium`).

- When SQL contains `ORDER BY COUNT (...) LIMIT ...`, rewrite the query to explicitly express the most or least intent for better intent alignment (**EX3**).

- Drop `GROUP BY` phrase if the column grouped by appears in `SELECT` and attach `EACH` to the specific column if the query does not express the most/least intent (see `GROUP` dropped in **EX3** - **EX4** but not **EX2**). This aims to distinguish SQLs with `GROUP BY` and `SELECT` on the same column from those without `SELECT`.

# 4 Experiments

We conduct experiment on the challenging Spider benchmark [Yu et al., 2018b], which contains various complex SQL statements and realistic cross-database evaluation setting. We demonstrate the effectiveness of our data synthesis framework from both text-to-SQL and SQL-to-text.

**Spider Benchmark**   Spider [Yu et al., 2018b] is a large-scale text-to-SQL dataset, it has 10,181 annotated questions, 5693 unique complex SQLs and 200 databases with multiple tables. It also contains Text2SQL datasets from previous work, which are compiled as **train-others**. The **train/train-others/dev/test** sets contain 7000/1659/1034/2147 examples and 140/6/20/40 databases, respectively. Since Spider test set is not publicly available, we use **dev** set for evaluation and **train-others** for checkpoint selection.

**Text-to-SQL Parser and SQL-to-Text Generator**   We use T5-3B [Raffel et al., 2020] as our base parser, since previous work [Shaw et al., 2021] has shown that T5-3B can achieve competitive performance for Text-to-SQL semantic parsing. We also apply PICARD [Scholak et al., 2021], a constraint decoding method, to further improve the syntactic correctness of the generated SQLs. We finetune a T5-large model on Spider training set for both SQL-to-text generator and IR-to-text generator, the best checkpoint is selected with the highest BLEU score on **train-others**.

**Configurations**   We train T5 with Adafactor and learning rate of 1e-4, and use batch size 2050 and 64 for T5-3B and T5-Large, respectively. Our experiments are based on NVIDIA A100-SXM4-40GB GPUs, we use beam size 5 and top-2 predictions for PICARD decoding.

## 4.1 Spider Results and Analysis

The overall results[3] are shown in Table 4. We can see that our synthetic data can further improve the state-of-the-art model and achieve the best results on Spider development set[4], including both exact set match and execution accuracy. Specifically, we have 4.4 points of EM score improvement on top of T5-3B model, while previous work [Wu et al., 2021, Wang et al., 2021] has marginal gain or even hurt the performance, demonstrating the effectiveness of our proposed method.

---

[3]Some models do not predict cell values or access to database content, we leave '-' for EX.

[4]Since the official test set is hidden, we have not received their evaluation results as of submission time

Table 4: Comparison of the top-performing text-to-SQL models in Spider leaderboard, as well as models trained with synthetic data. We report exact set match (EM) and execution accuracy (EX) for Spider dev set. † means T5-3B is trained with database content.

| Model | EM | EX | Model | EM | EX |
|---|---|---|---|---|---|
| DT-Fixup SQL-SP [Xu et al., 2021] | 75.0 | - | SmBoP + GraPPa [Rubin and Berant, 2021] | 69.5 | 71.1 |
| LGESQL + ELECTRA [Cao et al., 2021] | 75.1 | - | GAP + NatSQL [Gan et al., 2021a] | 73.7 | 75.0 |
| S2SQL + ELECTRA [Hui et al., 2022] | 76.4 | - | T5-3B† [Scholak et al., 2021] | 71.5 | 74.4 |
| DT-Fixup + Syn [Yang et al., 2021] | 76.4 | - | T5-3B† + Syn data (*ours*) | 74.5 | 78.6 |
| T5-3B [Shaw et al., 2021] | 70.0 | - | T5-3B† + PICARD [Scholak et al., 2021] | 75.5 | 79.3 |
| T5-3B + Syn data [Wu et al., 2021] | 69.1 | - | RASAT + PICARD [Qi et al., 2022] | 75.3 | 80.5 |
| T5-3B + Syn data [Wang et al., 2021] | 70.3 | - | T5-3B† + PICARD + Syn data (*ours*) | **76.1** | **81.4** |
| T5-3B + Syn data (*ours*) | 74.4 | - | | | |
| T5-3B + PICARD [Scholak et al., 2021] | 74.1 | - | | | |
| T5-3B + PICARD + Syn data (*ours*) | **76.9** | - | | | |

Table 5: Generated NLQ quality comparison between SQL$\rightarrow$ NLQ and SQL$\rightarrow$ IR$\rightarrow$ NLQ. We report BLEU [Papineni et al., 2002], ROUGE [Lin, 2004], and BERT [Zhang* et al., 2020] scores as evaluation metrics.

| Settings | BLEU | ROUGE-1 | ROUGE-2 | P-BERT | R-BERT |
|---|---|---|---|---|---|
| SQL$\rightarrow$ NLQ | 27.7 | 59.6 | 35.3 | 93.6 | 93.2 |
| SQL$\rightarrow$ IR$\rightarrow$ NLQ | 29.3 | 60.5 | 36.8 | 93.9 | 93.3 |

PICARD is an incremental parsing method for constraint decoding, which can reduce the syntax errors of language models for SQL generation. From Table 4, we see that T5-3B combined with PICARD and our synthetic data performs the best, implying the orthogonality of synthetic data augmentation and constraint decoding.

In order to understand the effectiveness of our proposed IR, we compared two generation paths in Spider benchmark: SQL→NLQ and SQL→IR→NLQ. As shown in Table 5, we can see IR helps the NLQ generation process and produces text closer to ground-truth NLQs.

### 4.2 Synthetic Data Efficiency

In this section, we study the efficiency of our synthetic data framework from two aspects:

*Few-shot setting: How much real data do we need to rely on before achieving acceptable performance?* Since annotating text-to-SQL dataset takes extremely high human effort, in practice, it's hard to create a large-scale corpus with a limited annotation budget. Table 6 presents the text-to-SQL semantic parsing results with different number of training examples. Interestingly, as training size decrease from 7K to 128, our synthetic data becomes more essential, and the performance gain increases from 4.4 points to 27.2 points. Even with only 512 training examples, our synthetic data can assist the T5-3B model to achieve ∼60% accuracy level.

Table 6: Text-to-SQL experiment with the few-shot setting. **# tmpl** and **# syn** represent the number of templates and synthetic data size, we report exact set match on the Spider dev set.

| Model | $f$-shot: | 128 | 256 | 512 | 1024 | full (7k) |
| | # tmpl | 68 | 116 | 205 | 318 | 746 |
| | # syn | 7839 | 10775 | 14457 | 17002 | 21851 |
|---|---|---|---|---|---|---|
| T5-3B | real only | 19.1 | 32.3 | 43.6 | 53.2 | 70.0 |
| | real + syn | 46.3 | 54.4 | 59.9 | 62.2 | 74.4 |

*Seen schema: How good of the synthetic data if we consider a broader coverage of database schema?* Since the cross-database evaluation setting presents generalization challenge for text-to-SQL parsers, our synthetic framework can potentially overcome this by utilizing more database schemas, or even ones that can implicitly cover the evaluation set. For example, we can take advantage of public schemas, such as WikiTables [Bhagavatula et al., 2015], GitTables [Hulsebos et al., 2021], WikiSQL [Zhong et al., 2017] and SQL tutorial websites, some of them are even schema source for Spider benchmark. In practice, people adapt model to a new domain by including targeting database schemas, we simply added 20 databases from dev set into our data synthesis pipeline, then trained text-to-SQL

parser with a T5-Large model, where we observed ~2 points of performance improvement compared to that with training schema only.

## 5  Conclusion

In this work, we proposed a data synthesis framework for text-to-SQL semantic parsing. After incorporating key relationships from schema, strong typing, schema-distance weighted column sampling and intermediate representation to bridge SQL → NLQ generation, we synthesized high-quality dataset that can further improve the state-of-the-art performance on Spider benchmark.

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
