# OpenReview forum: "Importance of Synthesizing High-quality Data for Text-to-SQL Parsing"
_NeurIPS.cc/2022/Workshop/SyntheticData4ML — Neurips 2022 SyntheticData4ML_

### Official Review · Reviewer_bHSK · 2022-10-15
**Interesting paper that appears relevant to the workshop**

**Rating:** 6
**Confidence:** 2

**Review:**

The paper presented an interesting method for synthesizing text-to-speech data using a combination of techniques including strong typing, weighted column sampling, and intermediate representation. Unsure if strong typing is a novel innovation. However, the results demonstrated respectable improvements in exact set match and execution accuracy over equivalent methods that didn't use these techniques.

---

### Official Review · Reviewer_WFGF · 2022-10-16
**Synthetic data generation for text-to-SQL**

**Rating:** 4
**Confidence:** 3

**Review:**

The paper discusses marginal benefits of synthetic data (by previous works) for text-to-SQL (Spider dataset) and identifies three issues of previous works (invalid grammar, arbitrary joins, language gap betweem NLQ and SQL). The paper proposes additional rules (strong typing, key relationship preservation, schema-distance-weighted column sampling, intermediate representation) which shows improvement on T5 (with and without PICARD) on Spider benchmark.

The writing of the manuscript needs improvement.

(1) The author may consider move section 2.1 to Experiment Section or provide context about section 2.1. PICARD needs to be introduced prior to this section. What are stage 1 and 2 in the experiment? Which benchmark was evaluated on (Spider?)?

(2) In Abstract, what is "pre-finetune"? Also Abtract mentions 2 shortcomings but section2.2 discusses 3 issues?

(3) Fig 2 better to be larger. The outputs of the proposed method and how the outputs combined with other dataset / model need to be illustrated more clearly.

(4) The relations between SQL synthesis and NLQ synthesis need to be described more clearly.

(5) The manuscript will benefit from proofread.

---

### Official Review · Reviewer_aeDc · 2022-10-19
**Review of "Importance of Synthesizing High-quality Data for Text-to-SQL Parsing"**

**Rating:** 7
**Confidence:** 3

**Review:**

This paper examines the effectiveness of data augmentation for downstream text-to-SQL tasks. The author identifies the issues with existing data augmentation methods and develops a practical solution that empirically leads to improved downstream performance.
Understanding when the synthetically augmented data leads to performance gain is an interesting topic.

The paper would benefit from minor revisions on grammar and punctuation.

---

### Official Review · Reviewer_LyET · 2022-10-20
**A few heuristic rules that lead to some improvements**

**Rating:** 6
**Confidence:** 4

**Review:**

The authors proposed a few heuristic rules when generating NLQ from synthetic SQL. The few rules introduced are 1) explicitly considering the column types (numerical vs categorical, etc.), 2) encouraging SQL to reference columns that are closer in distance (number of joins needed), 3) restrictions on foreign keys. When generating the intermediate representation, some additional heuristic rules are proposed, but they seem to be more on a case-by-case basis and seem to be less justified but are rather based on experience.

As for the experiments, I highly encourage the authors to test the rules on more datasets, as currently it is hard to tell whether those rules are engineered specifically for this Spider dataset. Additionally, in Table 6, f-shot is increased together with # syn and # tmpl. This makes it hard to judge where the improvements are coming from.

Overall, although the paper lacks in technical depth, the heuristic rules it proposes are simple to follow and appear to work well on the Spider dataset, so I recommend acceptance.

---

### Meta-Review · Area_Chair_xmH4 · 2022-10-19

**Recommendation:** Accept